# Impact of Colorectal Cancer Sidedness and Location on Therapy and Clinical Outcomes: Role of Blood-Based Biopsy for Personalized Treatment

**DOI:** 10.3390/jpm13071114

**Published:** 2023-07-10

**Authors:** Sasha Waldstein, Marianne Spengler, Iryna V. Pinchuk, Nelson S. Yee

**Affiliations:** 1Division of Hematology-Oncology, Department of Medicine, Penn State Health Milton S. Hershey Medical Center, Hershey, PA 17033, USA; sasha.waldstein@gmail.com (S.W.); mspengler@wellspan.org (M.S.); 2Vancouver Clinic, Vancouver, WA 98664, USA; 3Wellspan Medical Oncology & Hematology, Lebanon, PA 17042, USA; 4Division of Gastroenterology and Hepatology, Department of Medicine, Penn State College of Medicine, Cancer Control Program, Penn State Cancer Institute, Hershey, PA 17033, USA; ivp5097@psu.edu; 5Division of Hematology-Oncology, Department of Medicine, Penn State Health Milton S. Hershey Medical Center, Next-Generation Therapies Program, Penn State Cancer Institute, Hershey, PA 17033, USA

**Keywords:** colorectal cancer, subsites, molecular profiling, microbiota, liquid biopsy, personalized treatment, sidedness

## Abstract

Colorectal cancer is one of the most common malignant diseases in the United States and worldwide, and it remains among the top three causes of cancer-related death. A new understanding of molecular characteristics has changed the profile of colorectal cancer and its treatment. Even controlling for known mutational differences, tumor side of origin has emerged as an independent prognostic factor, and one that impacts response to therapy. Left- and right-sided colon cancers differ in a number of key ways, including histology, mutational profile, carcinogenesis pathways, and microbiomes. Moreover, the frequency of certain molecular features gradually changes from the ascending colon to rectum. These, as well as features yet to be identified, are likely responsible for the ongoing role of tumor sidedness and colorectal subsites in treatment response and prognosis. Along with tumor molecular profiling, blood-based biopsy enables the identification of targetable mutations and predictive biomarkers of treatment response. With the application of known tumor characteristics including sidedness and subsites as well as the utilization of blood-based biopsy, along with the development of biomarkers and targeted therapies, the field of colorectal cancer continues to evolve towards the personalized management of a heterogeneous cancer.

## 1. Introduction

In recent years, the differences between tumors originating from the right versus left side of the colon have been a focus of clinical investigation and biomedical research. Patients with right-sided colon cancer typically present with gastrointestinal bleeding and iron-deficiency anemia, whereas left-sided colon cancer usually manifests as colicky abdominal pain with changes in bowel habits and intestinal obstruction. It has become apparent that prognosis differs depending on tumor sidedness, and that response to therapy is impacted by tumor site of origin. An improved understanding of these unique differences has created new strategies for targeted therapy. The right, or proximal, colon is most often defined as the cecum, ascending colon, and the proximal two-thirds of the transverse colon. The left, or distal, colon includes the last one-third of the transverse colon, the descending colon, the sigmoid colon, and the rectum. The right colon originates from the midgut and is supplied by the superior mesenteric artery, whereas the left colon originates from the hindgut and is supplied by the inferior mesenteric artery [1,2]. In addition to basic anatomy and embryology, the differences span a wide range of characteristics, including epidemiologic features and morphology, as well as carcinogenic pathways, molecular mutations, the microbiome, and predisposition by antibiotics.

Colorectal cancer (CRC) is the third most common cancer worldwide and the fourth in the United States [3,4]. Based on the Surveillance, Epidemiology, and End Results (SEER) Program, there has been an overall downward trend in new CRC cases, with an annual decreased incidence of 2.3% on average. This downward trend is evident for patients older than the age of 55. However, there has been a clear increase in incidence in younger populations, starting from the age of 20 [5]. This early-onset CRC predominately originates from the left side of the colon [6]. Regardless of family history, and after excluding inherited syndromes and inflammatory bowel disease, the Cleveland Clinic identified that 83% of early-onset CRC in the study population originated from the left colon. This was similarly shown in a Mayo Clinic cohort [7]. Right-sided tumors are more common in older patients, and generally have a slight female predominance. The underlying causes for these epidemiologic trends remain largely unknown, but it is apparent that the differences in CRC, in terms of sidedness, are present even at this population level. With a growing understanding of the unique molecular and predisposing properties of right- versus left-sided CRC, these population trends may begin to be explained. The clinical features of right- vs. left-sided CRC are summarized in Figure 1.

In this article, we review the clinically important aspects of CRC relating to tumor sidedness, with updated data from the published literature. First, we provide an overview of the differences in genetic mutations and microbiota between right- and left-sided CRC. Special emphasis is placed on the difference in molecular profiles and microbiota that are relevant to targeted therapeutics and clinical outcomes. The prognosis of CRC at different stages, with relevance to tumor sidedness, is presented, as well as current treatment and investigational approaches. The emerging application of blood-based biopsy for precision treatment of right- versus left-sided CRC is highlighted.

## 2. Differences in Morphology and Genetic Mutations

There are striking differences in morphology and mutational status with regard to the sidedness of CRC. Right-sided CRCs are more commonly mucinous adenocarcinomas following a sessile serrated pathway, with a flat morphology and highly immunogenic component with increased T cell infiltration. In contrast, the left-sided CRC is often tubular villous, polypoid, and with a low immunogenic component [8]. In addition to these morphological characteristics, there is heterogeneity in mutational burden.

Studies have demonstrated that left-sided tumors are more likely to have mutations affecting *APC* and *TP53* through chromosomal instability, the most common pathway for sporadic CRC [9,10,11]. Left-sided tumors also have higher rates of *EGFR* mutation and *HER2* amplification [9]. On the other hand, a high degree of microsatellite instability (MSI-high), responsible for 10–15% of sporadic CRC, is present in about 22% of right-sided tumors as compared to 5% on the left. The CpG island methylator phenotype (CIMP-high) is also more common on the right, as are mutations of *PIK3CA* and *PTEN* [10]. Finally, *BRAF* mutations are present in 25% of right-sided CRC, as compared to 10% of left-sided cancers [10]. These tumor molecular profiles were also consistent in adolescent and young adult CRC, with higher mutation rates of *BRAF*, *KRAS,* and *PIK3CA* in right-sided tumors, as well as DNA mismatch repair gene mutations [12]. The morphological and genetic mutational features of right- and left-sided CRC are summarized in Figure 2.

However, these molecular features do not abruptly change at splenic flexure as demonstrated in a study involving 1443 CRC cases in 2 prospective cohorts [13]. The findings of this study indicate that the frequency of CIMP-high, MSI-high, and *BRAF* mutation gradually increases from the rectum to ascending colon. Interestingly, this trend in the changing frequencies of CIMP-high, MSI-high, and *BRAF* mutations as observed along the rectum to ascending colon is not displayed in cecal carcinomas, which exhibit a high frequency of *KRAS* mutations [13]. These findings are extended in a study of CRC from 1876 patients demonstrating that the prevalence of mutations differs among tumors located on the same side of the colon [14]. Specifically, the frequency of *BRAFV600* mutations increases from the cecum to the hepatic flexure. From the right colon to left colon, the rates of *TP53* and *APC* mutations decrease. There are variations in the MAP kinase and PIK3CA-mTOR-AKT pathways among tumors located within the same side of the colon. The mutational profiles of tumors in the transverse colon are significantly different from those in the right colon, but not tumors in the left colon [14]. Variations of tumor genetic alterations in different locations along the colorectum are clearly important in disease biology, and they also have important implications in patient prognosis and treatment responses.

## 3. Differences in the Gut Microbiome

The complex role of the microbiome in the pathogenesis of CRC, including the relation to tumor sidedness, has been extensively studied and reviewed [15,16]. Dysbiosis of the gut microbiome can contribute to the initiation and progression of malignant neoplasms, particularly CRC. The microbiota may potentiate malignant transformation through several mechanisms. These include the production of carcinogenic toxins, modulation of inflammatory pathways, dysregulation of epithelial signaling pathways, and DNA damage-driven genomic instability [16].

The composition of the microbiota has been found to have a modest gradient and variability across the colon, such as the relative abundance of *Fusobacterium* found in left-sided tumors [17]. Bacterial biofilms have been found to be present in 89% of right-sided CRC as compared to 12% of those on the left. Those patients with biofilms identified in their tumors also had biofilms present in their normal colonic mucosa [18]. It has been hypothesized that the increased propensity for cancer development in the setting of biofilms is associated with high immunogenicity, and there are complex interactions among the microorganisms and carcinogenic pathways [19].

Relating to the microbiome, the use of antibiotics for treating infection has been linked to varied effects on the risk of CRC [16,20,21]. The associations between antibiotic use and risk of CRC by tumor site have been examined [20,22]. Quinolones and sulfonamides and/or trimethoprims were significantly associated with an increased risk of cancer in the proximal colon but not the distal colon. On the other hand, nitrofurantoins, macrolides and/or lincosamides, and metronidazoles and/or tinidazoles were significantly associated with decreased risk of rectal cancer [22].

These lines of evidence indicate distinct microbiota in right- and left-sided CRC, suggesting a tumor site-specific effect of the gut microbiome in colorectal carcinogenesis. The understanding of the role of the microbiome in CRC will continue to expand, and these data highlight the need to continue to study CRC as a diverse entity that changes in nature along the course of the colorectum.

## 4. Prognosis and Tumor Recurrence

Though there is heterogeneity within and between the right- and left-sided CRC, there is substantial evidence that tumor sidedness represents a surrogate for known mutations, and those yet to be identified, that relate to prognosis. Right-sided CRC has many adverse features, such as higher rates of *BRAF* mutations, sessile serrated pathways of carcinogenesis, and MSI-high (MSI-H) disease. A meta-analysis demonstrated an absolute 19% reduced risk of death in left- as compared to right-sided colon cancer [23]. Importantly, this difference in mortality was not explained by race, cancer stage, chemotherapy, or year of study. Other studies have also shown that the relatively poor prognosis of right-sided metastatic CRC (mCRC) is independent of disease burden and known mutational status [24]. This reinforces that tumor sidedness serves as a proxy for additional prognostic factors that have yet to be identified. Considering the distinct profiles of right- versus left-sided mCRC, studies that stratify treatment by tumor site of origin have been essential to determining appropriate treatment.

The impact of tumor sidedness on clinical outcomes has also been investigated in localized colon cancer. In a retrospective study of 1632 patients in Korea, of which 15.8% had stage I, 36.3% stage II, and 47.9% stage III disease, those with right-sided colon cancer were found to have significantly increased risk of locoregional tumor recurrence [25]. The time to locoregional recurrence was significantly decreased in patients with right-sided colon cancer as compared to those with left-sided tumor, with a hazard ratio (HR) of 2.35 and *p* value < 0.001. The rate of 5-year locoregional recurrence was higher in patients with right-sided colon cancer (8.5%) than those with left-sided tumors. Other analyses of only stage I and stage II disease have found improved survival in those patients with right-sided tumors, potentially at least somewhat related to proportionally increased rates of MSI-H disease in that cohort which has been associated with a better prognosis in this early-stage disease [26,27].

In a retrospective study using the National Cancer Database, the benefit of adjuvant chemotherapy in patients with surgically resected stage II colon cancer, including the impact of tumor sidedness, was investigated [27]. Regardless of high-risk features, such as pathological tumor (T) stage 4 status and fewer than 12 lymph nodes examined, adjuvant chemotherapy was associated with improved overall survival (OS). In the sub-group analysis, patients with left-sided colon cancer received adjuvant chemotherapy more often (25.6%) than patients with right-sided tumors (17.9%) or transverse colon cancer (20.4%). In those that did not receive adjuvant chemotherapy, right-sided tumors had better OS outcomes as compared to those on the left, with an adjusted HR of 0.92. With the use of adjuvant chemotherapy, the 5-year survival rates were the same, and the difference was negated. This further helps delineate within this population those that would most benefit from adjuvant therapy. However, the impact of tumor sidedness on survival in patients with stage II colon cancer receiving adjuvant chemotherapy will need to be confirmed in prospective clinical studies.

The impact of tumor sidedness on survival in patients with stage III CRC seems to follow similar trends to that for mCRC. A study based on SEER data found that, although stage I–II right-sided disease had a lower risk of mortality than the left (left colon HR 1.09, rectum HR 1.36), there was a higher risk of mortality in right-sided stage III as well as stage IV disease, with a *p* value < 0.001 for each. In studies of patients with stage III disease, despite similar use of adjuvant chemotherapy, those with right-sided CRC had higher recurrence rates and decreased OS [28,29]. In one analysis, HR for right-sided CRC was 1.78 for recurrence, and right sidedness was an independent predictor of peritoneal recurrence specifically [28]. Therefore, although left- and right-sided CRC seem to have similar survival outcomes with adjuvant therapy in stage II disease, those patients with stage III disease and right-sided CRCs still have worse clinical outcomes than their left-sided counterparts.

The prognostic impact of the primary tumor sidedness in mCRC has also been investigated. In this study, the association between primary tumor sidedness and overall survival in patients with mCRC receiving first-line chemotherapy with or without bevacizumab in three independent cohorts was evaluated [30]. The results of this study indicate that patients with left-sided colon tumor had superior OS as compared with those with right-sided tumor, and there is no dependent relationship with the efficacy of bevacizumab. In agreement with this finding, another study demonstrated that the five-year OS rate of patients with left-sided colon cancer is superior to those with right-sided tumor [31]. Moreover, this difference in survival rate is independent of the *KRAS* mutational status and treatment with a biological agent (epidermal growth factor receptor inhibitor or bevacizumab). These data support the primary tumor sidedness as an important prognostic factor in patients with mCRC.

Moreover, the tumor location subsites of CRC have prognostic significance. In a study of 1876 patients with CRC with a median follow-up of 46.5 months, as compared to rectal tumors, the hazard ratios for OS in tumors located in the cecum, ascending colon, hepatic flexure, and transverse colon are 1.69, 1.72, 1.98, and 1.38, respectively [14]. In a recent study utilizing a consortium dataset of 13,101 patients with CRC, colon-cancer-specific survival was analyzed in relation to tumor location and molecular features [32]. A significant trend was demonstrated for improved survival in patients with cancer located in the cecum towards the sigmoid colon. For tumors with non-MSI-high, a significant trend for improved survival was shown in sigmoid colon with a hazard ratio of 0.80 relative to the cecum. An inverse trend was exhibited for tumors with MSI-high, and the lowest survival for tumors located in the sigmoid colon with a hazard ratio of 2.13 [32].

## 5. Impact of Tumor Sidedness on Treatment and Clinical Outcomes in Metastatic CRC

As knowledge of the distinguishing characteristics of left- and right-sided mCRC emerged, core studies of the treatment for mCRC were retrospectively analyzed by tumor side of origin. Chemotherapy with FOLFOX (5-fluorouracil, leucovorin, oxaliplatin) or FOLFIRI (5-fluorouracil, leucovorin, irinotecan) has been the mainstay of treatment for mCRC. The addition of bevacizumab, a vascular endothelial growth factor (VEGF) inhibitor, or anti-epidermal growth factor receptor (EGFR) therapy, with either cetuximab or panitumumab, have been studied and have shown benefit. Bevacizumab was originally shown to have an OS benefit when added to chemotherapy for mCRC, with a median OS of 20.3 months as compared to 15.6 months with chemotherapy alone, with *p* < 0.001 [33]. A retrospective analysis of that population and others treated with this therapy showed that the benefit of bevacizumab with chemotherapy was independent of primary tumor sidedness [34]. 

Anti-EGFR therapy has a more disparate effect. The CRYSTAL study showed improved progression-free survival (PFS) of cetuximab in addition to FOLFIRI as compared to chemotherapy alone on initial analysis. Subsequent analysis of those patients with *KRAS* wild-type tumors showed a similar result, with a median OS of 23.5 months as compared to 20 months with chemotherapy alone (*p* = 0.0093) [35]. The PRIME study similarly showed the benefit of panitumumab in addition to chemotherapy in this population [36]. However, on retrospective stratified analysis of these studies, a more varied response was observed. For tumors with a left-sided origin, the benefit of cetuximab and panitumumab was similar to results reported in the original studies. However, when the right-sided tumor population was analyzed, there was no difference in survival between chemotherapy alone and the addition of cetuximab. After stratification by sidedness, both in the CRYSTAL and PRIME studies, there was found to be no OS benefit from the addition of an EGFR inhibitor to chemotherapy for mCRC of right-sided origin. [37,38]. Based on these data, anti-EGFR therapy was determined not to be beneficial in right-sided mCRC, but it remains an option for those with left-sided cancers. 

Subsequent studies directly compared bevacizumab to anti-EGFR therapy, and similarly showed critical differences in treatment effect once the studied population was stratified by tumor sidedness. The FIRE-3, PEAK, and CALGB/SWOG 80,405 studies each evaluated chemotherapy plus bevacizumab or anti-EGFR therapy for *KRAS* wild-type mCRC. In FIRE-3 as well as PEAK, an OS benefit of anti-EGFR therapy, with cetuximab or panitumumab, respectively, over bevacizumab was demonstrated. This was not confirmed in the larger CALGB 80,405 study, which showed a median OS of about 30 months with both treatment regimens [39,40,41]. Each of these studies was later retrospectively stratified by tumor sidedness. The population of patients in the FIRE-3 trial with tumors originating from the left side of the colon had similar results as before stratification, with a median OS of 38.3 months with the addition of cetuximab, as compared to 28 months with bevacizumab, to FOLFIRI (*p* = 0.002) [37]. On analysis of right-sided tumors, there was no statistically significant difference in median OS between bevacizumab and cetuximab. Stratification of the PEAK trial by tumor sidedness demonstrated similar results [38]. When the same analysis was performed on the CALGB/SWOG 80,405 population, there was again an OS benefit with either bevacizumab or cetuximab for patients with left-sided tumors. However, right-sided tumors of origin appeared to do worse with the addition of cetuximab to chemotherapy as compared to bevacizumab, with a striking difference in median OS of 18.4 months as compared to 34.4 months, respectively [42]. An additional study of the California Cancer Registry supported this finding, with right-sided mCRC having an OS HR of 1.31 for cetuximab and chemotherapy as compared to HR of 0.93 for bevacizumab plus chemotherapy [43]. These subgroup analyses further support the need to address sidedness within trials of CRC, as this has changed the landscape for appropriate treatment recommendations.

Though the retrospective nature of these evaluations limits study design and conclusions, a convincing pattern, repeated in multiple studies, has emerged in patients treated with anti-EGFR antibodies beyond first-line treatment. In chemotherapy-refractory mCRC with primary left-sided tumor harboring wild-type *KRAS*, cetuximab significantly improved PFS as compared to best supportive care (5.4 months vs. 1.8 months, respectively, *p* < 0.001), but not in those with primary right-sided tumor (1.9 months vs. 1.9 months, respectively, *p* = 0.26) [44]. In agreement with this finding, for patients with either untreated and previously treated mCRC harboring wild-type *RAS* and *BRAF*, cetuximab either alone or in combination with irinotecan (if refractory to previous irinotecan) produced a disease control rate of 80% (partial response 40.7%, stable disease 39%) in patients with left-sided tumors of origin as compared to right-sided tumor (partial response 0%, stable disease 15.4%) [45]. This exemplifies the way in which primary tumor sidedness greatly affects response to treatment, even when known mutational abnormalities are controlled for. As a result of these studies, the current guidelines recommend EGFR inhibition only for left-sided tumors of origin with wild-type *RAS* and *BRAF*, whereas bevacizumab is indicated regardless of primary tumor sidedness.

The intensity of chemotherapy has also been studied in mCRC, and differences in treatment effect have again been shown with regard to the tumor site of origin. The TRIBE study evaluated the effect of FOLFOXIRI plus bevacizumab as compared to FOLFIRI plus bevacizumab [46]. PFS was found to be improved in the FOLFOXIRI group, and an updated OS analysis of the intention-to-treat population showed a median of 29.8 months with FOLFOXIRI as compared to 25.8 months with FOLFIRI, and the difference reached statistical significance [47]. This study population was subsequently stratified by the side of tumor origin. There was no difference in OS for left-sided tumors that received FOLFIRI vs. FOLFOXIRI plus bevacizumab, but there was an OS HR of 0.56 for right-sided tumors with the more intensive chemotherapy [48]. When *RAS* and *BRAF* WT-only tumors were evaluated, the HR was 0.5 for right-sided tumors as compared to HR of 0.88 for left-sided tumors, which was not statistically significant. The number of patients in this analysis was small, and there were increased toxicities with the intensive therapy. Though the retrospective nature of this study limits its conclusions, especially given that right-sided tumors are less common and therefore relatively less represented, it provides further evidence for the consideration of sidedness in guiding the initial intensity of therapy and the need to address sidedness in CRC treatment approaches and research. The clinical outcomes of these mCRC treatments, with regard to tumor sidedness, are summarized in Table 1.

## 6. Molecular Target-Based Treatment

Stratifying treatment of mCRC by tumor sidedness demonstrated how the primary site of tumor origin, a proxy for differences in tumor characteristics that are either known or yet to be identified, can make a significant impact on clinical outcomes. Through improved understanding of these differences and the pursuit of targeted approaches, new therapeutic advances have been made in the field of mCRC. Right-sided tumors, as discussed previously, have poor prognosis which, at least in part, is due to key differences in mutational status, apparent resistance to anti-EGFR therapy, and increased rates of MSI. There is therefore an opportunity to individual treatment by targeting these features to improve outcomes, with initial trials summarized in Table 2.

Through molecular testing and application of targeted therapy, new treatment options have been made available to patients with mCRC. The BEACON trial investigated targeted therapy of *BRAF V600E* mutated mCRC, more common in right-sided CRC, with either triplet therapy (encorafenib, binimetinib, and cetuximab), doublet therapy (encorafenib and cetuximab), or the investigator’s choice of either cetuximab plus irinotecan or FOLFIRI. Inhibition of *BRAF* with encorafenib and *EGFR* with cetuximab, with or without additional *MEK* inhibition by binimetinib, addresses the way in which single agent *EGFR* or *BRAF* inhibition are bypassed by the tumors for sustained cell growth and proliferation. Initial reports showed increased OS with both triplet and doublet regimens as compared to the control [49]. Almost two thirds of patients had right-sided tumors in origin, consistent with known patterns of *BRAF* mutational prevalence along the colon. An updated analysis of these data showed no difference between the doublet versus triplet regimen in terms of OS, with median of 9.3 months, but with increased side effect profile of additional MEK inhibition by binimetinib. Therefore, the doublet therapy has been FDA approved in this *BRAF V600E* mutated population [50].

Additional studies of mutational targeting have addressed *HER2* amplified tumors, which account for approximately 2–4% of all, predominately left-sided, mCRC. In both MyPathWay and HERACLES, treatment was aimed against HER2. In MyPathWay, a phase 2 trial, patients with treatment-refractory mCRC, with confirmed HER2 amplification, were administered pertuzumab and trastuzumab. An objective response was seen in 32% of patients, of which 57 were enrolled at the time of initial publication [51]. HERACLES also is a phase 2 trial for patients with HER2 amplified mCRC refractory to standard frontline treatments. These patients were administered trastuzumab and lapatinib. Fairly similar to the outcomes from MyPathWay, 30% of patients had an objective response, of the 27 patients initially enrolled [52]. This provides an additional treatment option for patients with HER2 amplification and highlights the need for the identification of such patients to offer this subsequent treatment.

Immunotherapy has also gained an increasing role in mCRC, as there has been heightened recognition of the prevalence of MSI-H disease, especially among the challenging right-sided mCRC population. Checkmate–142 was a phase II trial of low dose ipilimumab and nivolumab for patients with MSI-H disease. As first-line therapy, updated analysis showed an overall response rate of 60%, with median duration of response, PFS, and OS not yet reached after 19.9 months of median follow-up [53,54]. In this study, 55% of patients had right-sided colon cancer, with an additional 13% originating from the transverse colon, consistent with known differences in carcinogenesis pathways along the colon. Pembrolizumab has also been studied for first-line treatment of mismatch repair deficient tumors. The phase III Keynote-177 study showed a 48% PFS with pembrolizumab at 24 months as compared to 19% with chemotherapy. Median duration of response was not yet reached [55]. As a result of this study, pembrolizumab has been approved for first line treatment of MSI-H mCRC. The results for clinical trials using these targeted approaches, based on mutational status, for the treatment of mCRC are summarized in Table 2. 

These molecularly targeted approaches have advanced the field of mCRC by individualizing therapy rather than treating all mCRC cases as a homogenous entity. Given the effectiveness of this targeted approach, methods to further characterize cancers, as well as monitor response and resistance patterns during treatment, are essential. Liquid biopsy holds great promise in filling this role and allowing for patient-tailored treatment. 

## 7. Blood-Based Biopsy for Personalized Treatment

While tissue biopsy is the standard of care for diagnosing and molecularly profiling cancer, liquid (primarily blood-based) biopsy has emerged as a valuable tool in precision oncology for many cancer types, including CRC [56,57]. It is a relatively non-invasive and time- and cost-efficient technology for molecular profiling at the time of diagnosis or recurrence of disease, monitoring of tumor response to treatment, identification of resistance mechanisms, and revealing tumor heterogeneity. Current guidelines for mCRC require molecular profiling for *RAS*, *BRAF*, *HER2*, and MSI/MMR (microsatellite instability/mismatch repair) for optimal therapy selection [58], regardless of whether the primary tumor is left- or right-sided, or even in which colorectal subsite the tumor is located. Tissue-based analysis of *RAS* and *BRAF* mutations prior to initiation of treatment is necessary because patients with *RAS* and/or *BRAF* mutated tumors do not respond to anti-EGFR therapy. Blood-based biopsy has emerged as a useful tool for determining circulating tumor cells and circulating tumor DNA (ctDNA), and it potentially plays a role in personalized treatment of CRC. In a recent study, ctDNA can help guide the use of adjuvant chemotherapy in patients with stage II colon cancer [59]. Whether a ctDNA-based approach can guide the personalized treatment of right- vs. left-sided localized CRC or mCRC remains to be investigated.

Characterization of circulating tumor cells (CTC) in plasma has enabled the identification of differences in cancer biology and metastatic potential between left- and right-sided colon cancers, both quantitatively and qualitatively. In a retrospective analysis, 84 patients with metastatic CRC were subdivided according to tumor sidedness [60]. Despite containing the lowest median number of cells, CTC in left-sided CRC patients had the highest prognostic impact in terms of time to progression, at 11.1 months in patients with positive CTC as compared to 25.6 months in those with negative CTC. This inferior outcome was explained by the phenotypic heterogeneity exhibited by CTC from distal tumors, mainly the mesenchymal phenotype which allows epithelial cancer cells to acquire properties that facilitate their dissemination from the primary tumor site. This is in contrast to CTC from right-sided colon cancer, which despite being found in higher numbers, had little prognostic impact owing to the discovery that 80% of these cells exhibited an apoptotic pattern in comparison to only 10% from left-sided colon and rectal cancer.

Right- and left-sided CRC have divergent gene expression and mutation profiles, and, in addition to the expanding role it already plays, liquid biopsy has the potential to identify new molecular drivers of disease and resistance for which targeted therapies can be developed, thereby improving the outcomes of this patient population. Using whole-exome sequencing of plasma cell-free DNA from patients with gastrointestinal cancer including CRC, potential advantages of cfDNA over tissue biopsy for identification of acquired resistance alterations with therapeutic implications were suggested [61]. Future investigation of liquid biopsy in right- vs. left-sided CRC in localized stages to guide adjuvant therapy as well as the metastatic setting for selection of treatment will be indicated.

## 8. Conclusions and Future Perspectives

Right- and left-sided CRC have known differences in morphology, molecular mutations, and microbiomes. These differences, and those yet to be identified, lead to a worse prognosis in advanced tumors originating from the right side as well as varied responses to treatment. This is exemplified by the retrospective analyses of key studies regarding the treatment of mCRC based on sidedness, which have highlighted this disparate response. While bevacizumab, in addition to chemotherapy, benefits mCRC outcomes regardless of primary tumor sidedness, anti-EGFR therapy leads to longer OS in left-sided tumors of origin but not right-sided cancers. Apparently, CRC is not a single uniform entity, but rather a diverse tumor population that needs to be treated with a targeted approach. Though some of these factors seem to be reflected by tumor sidedness and location, the differences seen in prognosis and treatment are likely a result of a more complex set of specific unique features yet to be identified. In particular, the impact of tumor location in colorectal subsites on treatment response in relation to tumor sidedness remains largely unexplored.

Liquid (blood-based biopsy) with analysis of ctDNA has been utilized in patients with various malignant diseases including CRC to identify genetic alterations and guide selection of therapy. In addition, determination of ctDNA has been used to monitor tumor response to treatment, detect minimal residual disease and tumor recurrence, and help elucidate resistance mechanism. However, the role of liquid biopsy for identifying therapeutic targets and predictive biomarkers for treatment with regard to right- vs. left-sided CRC as well as the tumor location in colorectal subsites remains to be determined.

The importance of stratifying clinical and translational studies based on tumor sidedness and location has become clear, as has the need for continued study of the features that distinguish right- versus left-sided mCRC as well as their location subsites and the way to detect those characteristics. With the identification of specific mutations, gene amplifications, and MSI-H status, the landscape of treatment for mCRC has expanded and become more targeted with new options available for the disease subtypes. With a greater understanding of these diverse tumor characteristics, and the ability to more readily identify those features by tumor molecular profiling and liquid biopsy, personalized treatment can be pursued to improve the clinical outcomes of patients with CRC.

## Figures and Tables

**Figure 1 jpm-13-01114-f001:**
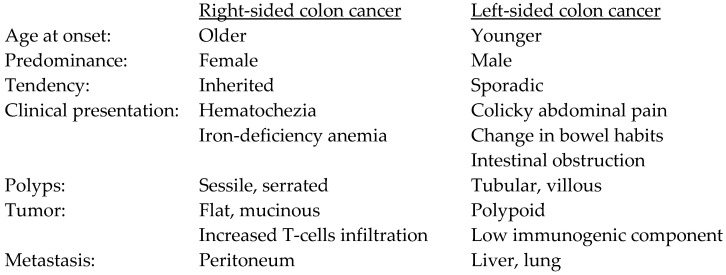
Clinical and pathological features of right- vs. left-sided colon cancer. The features listed in one side of the colorectum are relative to the sidedness of colorectum, and they are not distinctly different with regard to tumor sidedness.

**Figure 2 jpm-13-01114-f002:**
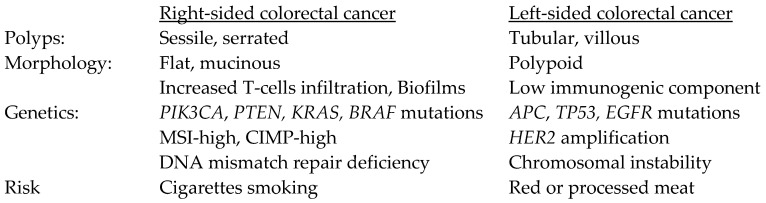
Morphological and genetic mutational features of right- and left-sided colon cancer. The listed molecular features may be more common in one side of the colorectum than the other side, though the frequency of molecular features gradually changes along colorectal subsites, rather than abruptly changing at splenic flexure. CIMP-high: CpG island methylator phenotype; MSI-high: high degree of microsatellite instability.

**Table 1 jpm-13-01114-t001:** Median overall survival (mOS) using different treatment regimens, for mCRC in regard to tumor sidedness.

Study	Intervention	mOS (Months) Entire Studied Population	mOS (Months) Right-Sided CRC	mOS (Months) Left-Sided CRC
Stratified Crystal Data [34]	Cetuximab + FOLFIRI vs. FOLFIRI	23.5 vs. 20 (* *p* = 0.0093)	18.5 vs. 15 (*p* = 0.76)	28.7 vs. 21.7 (* *p* = 0.002)
Stratified FIRE-3 Data [34]	Cetuximab + FOLFIRI vs. Bevacizumab + FOLFIRI	33.1 vs. 25.6 months (* *p* = 0.011)	18.3 vs. 23 (*p* = 0.28)	38.3 vs. 28 (* *p* = 0.002)
CALGB/SWOG 80405	Cetuximab + chemotherapy vs. Bevacizumab + chemotherapy	30 vs. 29 months (*p* = 0.08)	18.4 vs. 34.4 (* *p* = 0.03)	40.3 vs. 38.7 (* *p* = 0.04)
PRIME trial (*KRAS* WT)	Panitumumab + FOLFOX4 vs. FOLFOX4 alone	23.9 vs. 19.7 months (*p* = 0.17)	11.1 vs. 15.4 (*p* = 0.5398)	30.3 vs. 23.6 (* *p* = 0.0112)
Bevacizumab [30,31] (*RAS* mutation was not controlled for)	Bevacizumab + chemotherapy vs. chemotherapy alone	31 vs. 25.8 months *(p =* 0.054)	18.3 vs. 15.6 (*p* = 0.085)	23.5 vs. 20.8 (* *p =* 0.028)
TRIBE (*KRAS* and *BRAF* WT)	FOLFOXIRI + Bevacizumab vs. FOLFIRI + Bevacizumab	41.7 vs. 33.5 months (*p* = 0.52)	31.5 vs. 22.3	40 vs. 37.7
PEAK (*KRAS* WT)	Panitumumab + mFOLFOX6 vs. Bevacizumab + mFOLFOX6	36.9 vs. 28.9 months (*p* = 0.15)	17.5 vs. 21 (*p* = 0.3239)	43.4 vs. 32 (*p* = 0.3125)

FOLFIRI: 5-fluorouracil, leucovorin, irinotecan; FOLFOX: 5-fluorouracil, leucovorin, oxaliplatin. * *p* indicates statistical significance; WT: wild-type.

**Table 2 jpm-13-01114-t002:** Efficacy of targeted therapies in mCRC based on molecular mutation status.

Targeted Mutation	Trial	Intervention	Primary Endpoints
BRAF V600E	BEACON: treatment refractory mCRC	Encorafenib 300 mg PO daily and Cetuximab 400 mg/m^2^ IV initial dose, then 250 mg/m^2^ IV weekly vs. control (Cetuximab/Irinotecan or FOLFIRI)	OS 9.3 vs. 5.9 months HR 0.61 (0.48 vs. 0.77)
HER2 amplification	MyPathway: treatment refractory mCRC	Trastuzumab 8 mg/kg IV loading dose, followed by 6 mg/kg IV every 3 weeks plus Pertuzumab 840 mg IV loading dose followed by 420 mg IV every 3 weeks	ORR 32%
HER2 amplification	HERACLES: treatment refractory mCRC	Trastuzumab 4 mg/kg IV loading dose, followed by 2 mg/kg IV weekly + Lapatinib 1000 mg PO daily	ORR 30%
MSI-H	Checkmate-142: treatment naïve mCRC	Ipilimumab 1 mg/kg IV every 6 weeks and Nivolumab 3 mg/kg IV very 2 weeks	ORR 60% 12-month PFS 77% 12-month OS 83%
MSI-H	Keynote-177: treatment naïve mCRC	Pembrolizumab 200 mg IV every 3 weeks vs. chemotherapy	48% PFS at 24 months as compared to 19% with chemotherapy (* *p* = 0.0002)

HR: hazard ratio; mCRC: metastatic colorectal cancer; MSI-H: microsatellite instability-high; ORR: overall response rate; OS: overall survival; PFS: progression-free survival; (* *p* indicates statistical significance).

## Data Availability

Publicly available datasets were analyzed in this study.

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
