# Peer review of "Impact of Colorectal Cancer Sidedness and Location on Therapy and Clinical Outcomes: Role of Blood-Based Biopsy for Personalized Treatment"

_jpm, 2023, doi:10.3390/jpm13071114_

Round 1
Reviewer 1 Report
This is a nice article but lacks important discussion on the colorectal continuum model.
The authors should learn the colorectal continuum model (Yamauchi, Lochhead, et al. Gut 2012; Yamauchi, Morikawa, et al. Gut 2012). The authors discuss this model. It is much better and advanced than the dichotomy (or trichotomy) model used in many studies as well as the quoted obsolete reviews by Iacopetta (2002) and others. Subsite information certainly can give more personalised information on CRC than the trichotomy of right-side vs. left-side colon vs. rectum.
There are a number of studies which used this colorectal continuum model to examine colorectal tumour subsites, eg, Phipps et al. Cancer Epidemiol Biomarkers Prev 2012; Phipps et al. Cancer 2013; Jess et al. BMJ Open 2013; Mima et al. Clin Transl Gastroenterology 2016; Rosty et al. PLOS ONE 2013; Ugai et al. Am J Gastroenterol 2023; Ugai et al. J Gastroenterol 2023. The authors ignored these critical references, which also give an idea on how to analyse and interpret data on tumour location.
This colorectal continuum model has been supported by others, eg, Testa U et al. Mod Sci 2018; Li S et al. Biochim Biophys Acta Rev Cancer 2019; Caiazza F et al. Front Oncol 2015; Ito M et al. Int J Cancer 2014; Yamamoto H et al. Arch Toxicol 2015; Andrici J et al. Mod Pathol 2016; Lou Y et al. PLoS One 2015. The fact that all of these authors support the colorectal continuum model should be mentioned in this review.
Reviewer 2 Report
Thank you very much for giving me the chance to review this manuscript about current aspects and impact of colorectal cancer sideness. The manuscript is well designed and structured. The main focus is on onclogical targets. However, this is a very important topic and gets more and more complex. Congratulations on this work.
Author Response
We want to thank Reviewer #2 for the comments.